# A New Data Fusion Algorithm for Wireless Sensor Networks Inspired by Hesitant Fuzzy Entropy

**DOI:** 10.3390/s19040784

**Published:** 2019-02-14

**Authors:** Jiayao Wang, Olamide Timothy Tawose, Linhua Jiang, Dongfang Zhao

**Affiliations:** 1School of Optical-Electrical and Computer Engineering, University of Shanghai for Science and Technology, Shanghai 200093, China; wjiayao0203@163.com; 2Department of Computer Science and Engineering, University of Nevada, Reno, NV 89557, USA; otawose@nevada.unr.edu; 3Department of Computer Science, University of California, Davis, CA 95616, USA

**Keywords:** WSNs, hesitant fuzzy entropy, data fusion, energy consumption

## Abstract

The wireless sensor network (WSN) is mainly composed of a large number of sensor nodes that are equipped with limited energy and resources. Therefore, energy consumption in wireless sensor networks is one of the most challenging problems in practice. On the other hand, data fusion can effectively decrease data redundancy, reduce the amount of data transmission and energy consumption in the network, extend the network life cycle, improve the utilization of bandwidth, and thus overcome the bottleneck on energy and bandwidth consumption. This paper proposes a new data fusion algorithm based on Hesitant Fuzzy Entropy (DFHFE). The new algorithm aims to reduce the collection of repeated data on sensor nodes from the source, and strives to utilize the information provided by redundant data to improve the data reliability. Hesitant fuzzy entropy is exploited to fuse the original data from sensor nodes in the cluster at the sink node to obtain higher quality data and make local decisions on the events of interest. The sink nodes periodically send local decisions to the base station that aggregates the local decisions and makes the final judgment, in which process the burden for the base station to process all the data is significantly released. According to our experiments, the proposed data fusion algorithm greatly improves the robustness, accuracy, and real-time performance of the entire network. The simulation results demonstrate that the new algorithm is more efficient than the state-of-the-art in terms of both energy consumption and real-time performance.

## 1. Introduction

Wireless sensor networks (WSNs) are multi-hop self-organizing network systems formed by a large number of inexpensive wireless sensor nodes through wireless communication. Their purpose is to cooperatively perceive, collect, and process the information of perceived objects in the monitoring area and send it to the terminal. As an important part of Internet of Things (IoT), WSNs have broad application prospects in environmental monitoring, national defense, military, medical and health care, smart home, and many others. In most cases, sensor nodes are battery-powered and not rechargeable; moreover, they are usually deployed in environments that are inaccessible or difficult to reach and are expected to operate for months. Therefore, how to minimize those nodes’ energy consumption and maximize the network life cycle has recently attracted a lot of research interest.

In most applications, the sensor nodes are randomly deployed in a high-density target area. To keep the cost low, those sensor nodes that are responsible for collecting data are usually made from inexpensive parts. Therefore, when deployed in a variety of complex environments, such as mining fields and fish farms, these sensor nodes might become highly unreliable for data acquisition. To this end, it is unfeasible to only rely on a single sensor node to collect the data of a monitoring object. The state-of-the-art is to deploy multiple sensor nodes to monitor the same object to improve the reliability of the monitoring information. 

Monitoring the same object using multiple sensor nodes leads to the situation of receiving multiple relevant, possibly redundant, data of the same object. How to obtain accurate and reliable information from these redundant data is a key research question. Recent literature shows that energy consumption in WSN comes from two sources: computing and communication [1]. Since the ratio between computing- and communication-incurred energy consumption is about 1:3000, sensor nodes should focus on local simple data processing and reduce long-distance data transmission. In order to achieve this goal, sensor nodes must work in a coordinated manner. One possible approach is to organize WSN nodes in clusters, which forms a tighter interaction and collaboration between nodes while facilitating the reliability of the entire WSNs and saving network bandwidth. Moreover, data fusion within the cluster can effectively optimize redundancy, reduce the amount of data transmission and energy consumption in the network, extend the network life cycle, and improve bandwidth utilization. According to the fusion operation level, data fusion is divided into data-level fusion, feature-level fusion, and decision-level fusion [2]. Among them, decision-level data fusion is a process of the fusion of raw data collected by various sensor nodes, whose main purpose is to obtain high-quality information and make accurate decisions on events of interest based on the data collected from various sensor nodes.

In this paper, we propose a new data fusion algorithm at the decision level. The proposed algorithm is inspired by hesitant fuzzy entropy [3] to distribute the weight of the redundant data to extract meaningful and reliable data fusion to make local decisions. The algorithm gathers nodes’ local decisions and sends them to the base station to reach the final decision. Our algorithm can improve the reliability and accuracy of the network, reduce the transmission of the original data in the network, extend the entire network life cycle, and prevent network congestion.

In the next section, we will introduce the research status of data fusion in wireless sensor networks. The third section introduces the network model, energy model, and cluster formation of the network system. In the fourth section, a data fusion algorithm based on entropy of the hesitant fuzzy set is proposed. The fifth section demonstrates the accuracy and robustness of the system through simulation experiments. The sixth section concludes this paper.

## 2. Related works

In our prior work [4], we proposed a new topology for a three-tier WSN inspired by the Chinese Remainder Theorem to eliminate the conventional routing tables. The new topology reduced the network traffic and energy consumption, allowed for query parallelism, and incorporated a new protocol to effectively aggregate and deduplicate redundant data in a neighborhood cluster. In this paper, we have proposed a new data fusion algorithm inspired by hesitate fuzzy entropy within the clusters to reduce redundant data transfer and further improve the reliability of the network. Most common data fusion models are based on trees, clusters, particle swarm optimization, and so on. Yuan et al. [5] asserted that the path to the sink node determines where the data can be fused, thus influencing the efficiency of the aggregation process. The topology is constructed by descending the path traversing the tree, and data fusion is carried out at the same time. Chen et al. [6] opined that relevant data are fused by constructing an aggregation tree. Luo et al. [7] mainly discusses the problem of minimum energy reliable information collection in data fusion and proposes a practical distributed approximation algorithm for chain and tree topology, which improves the reliability of the data fusion tree. In [8], the minimum fusion steiner tree was proposed to conduct efficient data collection by aggregation in wireless sensor networks. A random algorithm was adopted to allow fusion points to be selected according to the data volume of nodes. The tree-based data fusion algorithm can indeed reduce data redundancy to a certain extent, but in the process of building the tree, more time and extra energy may be needed, resulting in the failure of the entire network system to respond to emergencies in a timely manner.

Su et al. [9] combined the fuzzy logic method with clustering and proposed a data fusion method to ensure fault tolerance between sensors and effectively improve the communication bandwidth available between network components. A clustering based fuzzy logic theory data fusion method was proposed in [10] to examine the influence of inaccurate WSN observation values and different fuzzy sets on the data fusion algorithm. Most of these cluster-based algorithms focus on improving the accuracy of data, but the improved accuracy is also relative, and the final accuracy remains to be discussed. Sung et al. [11] proposed an improved local optimization method to improve the measurement accuracy of multi-sensor data fusion in the Internet of things. Data aggregation can be expressed as multidimensional according to particle swarm optimization. The improved particle swarm optimization algorithm can find the minimum solution of the objective cost function in the multidimensional allocation topic, which is considered in the particle swarm initialization, crossover rule, and mutation rule. Through effective candidate measures, the optimal isolation degree can be found effectively, to narrow the search scope. Dhivya et al. [12] adopted the Cuckoo Based Particle Approach (CBPA) to optimize the network. Nodes are randomly deployed and organized into static clusters through Cuckoo Search (CS). After selecting the cluster head, the information is collected, marked, and transmitted to the base station using the generalized particle approximation algorithm. The generalized particle model algorithm (GPMA) transforms the network energy consumption problem into the dynamics and kinematics problems of many particles in the force field. Such data fusion based on particle swarm optimization cannot be used in scenarios with high real-time requirements, and it consumes more energy than the common data fusion algorithms.

At present, there are few applications of hesitant fuzzy entropy to wireless sensor network data fusion, but there are many applications of fuzzy theory to wireless sensor network data fusion. An example of such is [13], where all kinds of sensor signals in the cluster head are collected and a method based on fuzzy rules for fusion is used to improve the reliability and accuracy of perception information and reduce the false data packet rate. Also, in [14], a wireless sensor data fusion method based on fuzzy theory was proposed to improve the service quality of the wireless sensor network. This method can only distinguish the authenticity of collected data and hence reduces the burden of data processing at the base station. Korvin et al. [15] employed the use of fuzzy set theory to aggregate the data of multiple sensors at the cluster level, obtain higher quality information from the collected original data, and make accurate decisions on matters of interest. Similarly, [16] solved the network congestion problem based on the fuzzy strategy. This kind of algorithm greatly reduces the redundancy of data, but it does not consider the impact of such data fusion on the accuracy of data, resulting in data distortion.

In addition, some applications combine fuzzy theory with other theories to solve data fusion problems. Banerjee et al. [17] used an order weighted average operator and fuzzy measure to determine the weight of data from different sensors, and then local aggregation was carried out to solve data incompleteness and improve data security. In [18], an adaptive fuzzy logic algorithm, which is superior to the average fuzzy logic algorithm, was proposed to solve the inaccuracy of data fusion. Hao et al. [19] used fuzzy logic to make initial local decisions at the group head before uploading to the base station. The base station then used the uploaded local decisions to calculate the fusion support degree of clusters and conducted the final fusion of events. Zhai et al. [20] mainly adopted hesitant language preference relationships (HLPRs) to put forward correct opinions on different wireless sensor network schemes, and fuse uncertain data information to improve the credibility of the overall network. The state-of-the-art approaches as explained above can improve the reliability of information collection, reduce the transmission of redundant information, or be employed to make accurate decisions on events of interest through data fusion. This paper attempts to achieve all of these effectively with a new data fusion algorithm based on hesitant fuzzy entropy. 

## 3. System Model and Cluster Formation

This section mainly introduces several important models of the network system and the formation of clusters.

### 3.1. Network Model

In this paper, WSNs have the following characteristics: 1. The network system is divided into three types of nodes: a base station, several sink nodes, and a large number of sensor nodes; 2. the base station has enough energy and storage space, and a powerful computing capacity. It mainly collects local decisions of each sink node and conducts comprehensive analysis on them. Finally, it makes a reasonable prediction on the occurrence probability of current events in the whole monitoring area and sends the final prediction results to the terminal. In order to better illustrate the algorithm in this paper, we assume that the base station is stationary; 3. the Sink node location is fixed. It also has sufficient power and storage space, and a strong processing ability. Sink nodes act as cluster heads in the network and receive data periodically from their neighbor sensor nodes; 4. the monitoring area of all sink nodes can cover the whole monitoring area, and each sensor node has its own cluster head (i.e., sink node); 5. sensor nodes have limited energy and storage space and can perform simple calculations.

As shown in Figure 1, this network model is composed of a base station, several sink nodes, and a large number of sensor nodes. Sensor nodes have the following characteristics: 1. Sensor nodes are randomly distributed and sink nodes are distributed uniformly in the monitoring area, as far as possible; 2. the sensor nodes in the monitoring area are isomorphic. The initial energy $E_{0}$ of the sensor nodes is the same and limited, and the energy consumption of various sensor nodes in the simulation experiment is also the same; 3. each sensor node has a unique ID in the network and can only perceive one attribute signal; 4. the sensor node position is fixed once deployed and cannot be moved. When a sensor node sends out a packet, it puts the residual energy $E_{r}$ in the header file of the data packet to convey the information to the neighbor nodes; 5. the communication radius of all nodes is adjustable and can communicate directly with the parent sink node; 6. due to the high density of sensor nodes, there is a large amount of redundancy in the data collected by nodes, which requires data fusion processing; 7. the sensor node can perceive the distance from its neighbor nodes through RSSI [4].

### 3.2. Energy Model

In this paper, we assume that the base station and sink nodes have sufficient energy. Complex operations, such as data fusion, are operated from sink nodes. The energy consumption of those sensor nodes comes from three states: the idle state, the transmitting state, and the receiving state. The energy consumption of data processing is not considered for sensor nodes because they are simply redundant data deletion. The energy consumption in the idle state is negligible; the transmitting energy ($E_{TX}$) and the receiving energy ($E_{RX}$) of transmitting a n-bit message to d-distance away are calculated as follows:(1)$$E_{TX}\left(n,d\right)=n\cdot E_{elec}+n\cdot d^{k}\cdot \varepsilon_{amp}\;E_{RX}\left(n\right)=n\cdot E_{elec}$$
where $E_{elec}$ denotes the energy consumption of each transmitter or receiver, $\varepsilon_{amp}$ denotes the energy consumption of the amplifier by one square meter per bit, and k denotes the propagation attenuation index. In practice, the choice of k [4] usually ranges between 2 and 5, depending on the environment; if higher interference (e.g., high buildings, dense forests) is expected, a higher k value should be set. 

### 3.3. Cluster Formation

In this network system, we use the method of Chinese Remainder Theorem (CRT) for topology construction and fault-tolerant and low-power transmissions to transmit data [4]. The three-tier CRT coding system is built using a bottom-up approach. That is, we assume that the total number of sensor nodes is known; calculate the size of the storage nodes; and finally, determine the base station’s keyword *N*. Once *N*’s range is known, we employ a top-down approach to determine all the keywords for storage nodes, each of which further determines the keywords of its assigned sensor nodes. According to the CRT, *N* will be divided into *k* coprime integers $n_{1},n_{2},\cdots ,n_{k}$, where k represents the number of storage nodes stored in the network. Each storage node’s $n_{k}$, again according to the CRT, is decomposed into coprime integers $n_{kj}$, where j indicates the secondary-index assigned to each sensor node, These sensor nodes are the children of the sink node. According to the Chinese Remainder Theorem, each sink node has a unique identifiable two-dimensional coordinate and each sensor node has a unique identifiable three-dimensional data transmission for preparation.

In this paper, sink nodes are taken as cluster heads, and other nodes in the cluster are composed of child nodes with their communication range. When the network topology ends, the clustering ends. In order to facilitate the study of data fusion, cluster-head nodes have a sufficient capacity to act as sink nodes, so we do not need frequent elections of cluster-head nodes. The children sensor nodes in the cluster will periodically integrate the collected data and send it to their parent sink nodes. Each cluster must include all sensor nodes that can detect possible event attributes and the summation of the communication range of similar sensor nodes in each cluster such that the entire cluster can be covered. In general, the base station is mobile because it has a large area to manage and requires a separate protocol to regulate its motion. However, the purpose of this paper is to discuss data fusion, so it is necessary to assume that the base station is stationary to facilitate subsequent research and highlight the research focus. In this network system, the base station is able to communicate directly with all sink nodes and sensor nodes. Sink nodes can communicate with the base station directly or by means of multiple hops. Sink nodes can communicate with each other directly or through multiple hops, sink nodes and sensor nodes in the same cluster can communicate with each other, and sensor nodes within the same cluster can also communicate with each other. Hence, nodes can either communicate with each other directly or through multiple hops.

## 4. Data Fusion Algorithm Based on Hesitant Fuzzy Entropy and Illustration of an Example

In general, multiple attributes are required to evaluate whether an event will occur (such as whether a forest is on fire). In WSNs, in order to ensure the accuracy of monitoring data, the same attribute value will be monitored by multiple sensor nodes. The same monitoring values are different due to the interference of various reasons (such as noise interference, different distances from the event occurrence point, etc.). When the sink node receives these inconsistent data, it is worth discussing how to make a correct judgment on the attribute value. The theory of hesitant fuzzy sets can solve this problem precisely. The membership degree of an element that is allowed to belong to a set can be several possible values. In WSNs, multiple possible values of a property value correspond to multiple possible memberships of hesitant fuzzy sets. An accurate entropy weight model is constructed by using hesitant fuzzy entropy, and the corresponding weight of each sensor monitoring value of this attribute is obtained. Then, the final result of this attribute is obtained according to the information provided by the system in advance and periodically transmitted to the base station. As we know, the main energy consumption of sensor nodes is communication. If the sensor nodes send the data to the sink node as soon as they receive the data, the sensor nodes in the region with frequent changes in events will greatly shorten their service life due to frequent sending of monitoring values. Therefore, this paper proposes a data fusion algorithm based on hesitant fuzzy entropy to solve the above problems, and further illustrates the principle of the data fusion algorithm with the forest fire prediction system.

### 4.1. The Introduction of Hesitant Fuzzy Entropy

Information entropy represents the statistical characteristics of the whole information source and is a measure of the overall mean uncertainty. A hesitant fuzzy set describes the hesitant situation when people make a decision. In this section, we shall develop some entropy measures for hesitant fuzzy information and discuss their relationships.

**Definition** **1**[21]**.**
*Let X be a fixed set, and the hesitant fuzzy set on X is in terms of a function α that when applied to X returns a subset of [0,1], which can be represented as the following mathematical symbol:*
(2)A={<x,α(x)>|x∈X}
*where α(x) is a set of values in [0,1], denoting the possible membership degrees of the element x∈X to the set A. For convenience, Xia and Xu [22] named α(x) an HFE and H the set of all HFEs.*

It is noted that the number of values in different HFEs may be different, so let $l_{\alpha \left(x\right)}$ be the number of values in α(x). We arrange the elements in α(x) in increasing order, and let $\alpha {\left(x\right)}_{\sigma \left(i\right)}\;\left(i=1,2,\cdots ,l_{\alpha \left(x\right)}\right)$ be the ith smallest value in α(x). Given an HFE α, Torra and Narukawa [23] defined the complement set of α described as $\alpha^{c}=\cup_{\gamma \in \alpha }\left\{1-\gamma \right\}$. In this paper, to operate correctly, we assume that the HFEs α and β should have the same length l when we compare them. If there is only one value in α, we should extend it by repeating the value until it has the same length of β.

In the following, we give the axiomatic definition of entropy and cross-entropy for HFEs:

**Definition** **2**[24]**.**
*An entropy on HFE α real-valued function E:H→[0,1], satisfying the following axiomatic requirements:*E(α)=0, if and only if α=0 or α=1;$E\left(\alpha \right)=1,\;if\;and\;only\;if\;\alpha_{\sigma \left(i\right)}+\alpha_{\sigma \left(l-i+1\right)}=1,\;for\;i=1,2,\cdots ,l_{\alpha }$;$E\left(\alpha \right)\leq E\left(\beta \right),\;if\;\alpha_{\sigma \left(i\right)}\leq \beta_{\sigma \left(i\right)}\;for\;\beta_{\sigma \left(i\right)}+\beta_{\sigma \left(l-i+1\right)}\leq 1\;or\;\alpha_{\sigma \left(i\right)}\geq \beta_{\sigma \left(l-i+1\right)},for\;\beta_{\sigma \left(i\right)}+\beta_{\sigma \left(l-i+1\right)}\geq 1\;i=1,2,\cdots ,l$;$E\left(\alpha \right)=E\left(\alpha^{c}\right)$.
(3)$$E\left(\alpha \right)=-\frac{1}{l_{\alpha }\ln2}\overset{l_{\alpha }}{\underset{i=1}{\sum }}\left(\frac{\alpha_{\sigma \left(i\right)}+\alpha_{\sigma \left(l_{\alpha }-i+1\right)}}{2}\ln(\frac{\alpha_{\sigma \left(i\right)}+\alpha_{\sigma \left(l_{\alpha }-i+1\right)}}{2}\;\;+\frac{2-\alpha_{\sigma \left(i\right)}+\alpha_{\sigma \left(l_{\alpha }-i+1\right)}}{2}\ln\frac{2-\alpha_{\sigma \left(i\right)}+\alpha_{\sigma \left(l_{\alpha }-i+1\right)}}{2}\right)$$

**Definition** **3**[24]**.**
*Let α and β be two HFEs. Then, the cross-entropy C(α,β) of α and β should satisfy the following conditions:*
$$C_{A\left(\alpha ,\beta \right)}\geq 0;$$
$$C_{A\left(\alpha ,\beta \right)}=0\;if\;and\;only\;if\;\alpha_{\sigma \left(i\right)}=\beta_{\sigma \left(i\right)},\;i=1,2,\cdots ,l.$$
(4)$$\begin{array}{c} C_{A\left(\alpha ,\beta \right)}\\ =\frac{1}{lT}\sum_{i=1}^{l}(\frac{\left(1+q\alpha_{\sigma \left(i\right)}\right)\ln\left(1+q\alpha_{\sigma \left(i\right)}\right)+\left(1+q\beta_{\sigma \left(i\right)}\right)\ln\left(1+q\beta_{\sigma \left(i\right)}\right)}{2}\\ -\frac{2+q\alpha_{\sigma \left(i\right)}+q\beta_{\sigma \left(i\right)}}{2}\ln\frac{2+q\alpha_{\sigma \left(i\right)}+q\beta_{\sigma \left(i\right)}}{2}\\ +\frac{\left(1+q\left(1-\alpha_{\sigma \left(l-i+1\right)}\right)\ln(1+q\left(1-\alpha_{\sigma \left(l-i+1\right)}\right)\right)+\left(1+q\left(1-\beta_{\sigma \left(l-i+1\right)}\right)\ln(1+q\left(1-\beta_{\sigma \left(l-i+1\right)}\right)\right)}{2}\\ -\frac{2+q\left(1-\alpha_{\sigma \left(l-i+1\right)}+1-\beta_{\sigma \left(l-i+1\right)}\right)}{2}\ln\frac{2+q\left(1-\alpha_{\sigma \left(l-i+1\right)}+1-\beta_{\sigma \left(l-i+1\right)}\right)}{2}),\;q>0 \end{array}$$
*where*
T=(1+q)ln(1+q)−(2+q)(ln(2+q)−ln2),q>0.

**Theorem** **1**[24]**.**
*Let α be an HFE. Then, $E_{A}\left(\alpha \right)=1-C_{A}\left(\alpha ,\alpha^{c}\right)$ is an entropy formula for α.*
(5)$$E_{A}\left(\alpha \right)=1-C_{A}\left(\alpha ,\alpha^{c}\right)=1-\;\frac{2}{l_{\alpha }T}\sum_{i=1}^{l_{\alpha }}(\frac{\left(1+q\alpha_{\sigma \left(i\right)}\right)\ln\left(1+q\alpha_{\sigma \left(i\right)}\right)+\left(1+q\left(1-\alpha_{\sigma \left(l_{\alpha }-i+1\right)}\right)\right)\ln\left(1+q\left(1-\alpha_{\sigma \left(l_{\alpha }-i+1\right)}\right)\right)}{2}\;-\frac{2+q\alpha_{\sigma \left(i\right)}+q\left(1-\alpha_{\sigma \left(l_{\alpha }-i+1\right)}\right)}{2}\ln\frac{2+q\alpha_{\sigma \left(i\right)}+q\left(1-\alpha_{\sigma \left(l_{\alpha }-i+1\right)}\right)}{2}),\;q>0$$
*where T=(1+q)ln(1+q)−(2+q)(ln(2+q)−ln2),q>0.*

### 4.2. Data Fusion Model Construction

In this network system, sink nodes periodically deliver the latest data fusion message to the base station. In order to reduce the frequency of communication between sensor nodes and sink nodes, sensor nodes are required to periodically transmit the collected data to sink nodes. As long as the size of the cycle is controlled, the real-time performance of the whole network can be maintained, and the number of times sensor nodes communicate can be greatly reduced. In this paper, it is assumed that the period required for the sink node to transmit messages to the base station is T, and that the period for the sensor node to transmit messages to the sink node is t, so T=nt. Moreover, instead of every sensor node in the cluster passing messages to the sink node during every *t* time period, it can be divided into the following two situations: In the first case, an event occurs in the area monitored by the sensor node, which wakes it up, and the sensor node passes the collected data to the sink node; in the second case, during the process of event occurrence, the data collected by the sensor node changes, and it is different from the data transferred in the previous period t, so the data will be transferred again, otherwise it will not be transferred. This scheme can greatly reduce the communication times of sensor nodes, thus extending the life of sensor nodes and the network life cycle.

Because the data passed by the sensor nodes are the attribute value in a time period t, within this period of time t, the sensor nodes monitoring data value may be the same or will continue to change (for example, the temperature may rise or fall), so it may be the form of $D_{i}$ {$d_{1},d_{2},\cdots ,d_{n}$}, where n≥1 and $d_{i}$ represents the different values collected by the sensor node in time t. Hence, the question of how to get reasonable monitoring values from these related values arises. This paper uses hesitant fuzzy entropy to solve the data fusion problem in Algorithm 1. The specific data fusion steps are as follows:
The sink node integrates and standardizes the data collected within a period T;The sink node lists all possible data values $G_{j}$ of all attribute values $AD_{i}$, and generates the hesitant fuzzy set $\alpha_{ij}$ (i=1,2,⋯,m;j=1,2,⋯,n);The matrix of hesitant fuzzy entropy $E\left(\alpha_{ij}\right)$ of the data at this stage is calculated through Formula (5);In order to obtain the weight value $w_{i}$ of each set of sensor nodes, an accurate entropy weight model is constructed by the matrix of hesitant fuzzy entropy $E_{ij}$:(6)$$w_{i}=\frac{1-E_{i}}{m-\sum_{i=1}^{m}E_{i}}$$
where $E_{i}=\frac{1}{n}\sum_{j=1}^{n}E\left(\alpha_{ij}\right),i=1,2,\cdots ,m$.According to Formula (7), the data value of attribute $F_{j}$ is synthesized:(7)$${\bar{D}}_{F_{j}}=\frac{1}{n}\overset{n}{\underset{i=1}{\sum }}w_{i}D_{i}$$
where $D_{i}=\frac{1}{l_{\alpha }}\sum \alpha_{\sigma \left(i\right)},i=1,2,\cdots ,m.$According to Formula (8), the attribute value is converted into the possible probability *P* of the occurrence of events.
(8)$$P={\bar{w}}_{j}\overset{n}{\underset{j=1}{\sum }}\frac{{\bar{D}}_{F_{j}}}{D_{F_{j}}^{\prime }}$$

The ${\bar{w}}_{j}$ is set by the system in advance of the weight value of various parameters leading to the final event occuring, and $D_{F_{j}}^{\prime }$ represents the threshold value of attribute $F_{j}$ when this event occurs. When the value of property $F_{j}$, inversely proportional to the probability of an event happening, is $p_{F_{j}}=1-\frac{{\bar{D}}_{F_{j}}}{D_{F_{j}}^{\prime }}$, $D_{F_{j}}^{\prime }$ represents the threshold value of attribute $F_{j}$ when this event is impossible.

It is important to note that the sink nodes integrate the collected data as follows: At the end of the cycle, the sink node will send the collected data from the storage to the data processing unit for processing. The sensor node continues to receive the data in the next period, while the data processing unit merges the collected data. The sink node will calculate the interval values that may appear in each attribute value, divide the interval values into 10 parts, and assign the 10 intervals values of {0–0.1, 0.1–0.2, 0.2–0.3, 0.3–0.4, 0.4–0.5, 0.5–0.6, 0.6–0.7, 0.7–0.8, 0.8–0.9, 0.9–1.0} respectively. Therefore, the collected data will be assigned the corresponding interval value according to the size of its value, in order to complete the integration and standardization processing.

The following illustrates the whole process of network data fusion:

**Algorithm 1.** Data fusion process.1. When an event occurs, sensor nodes are awakened to perceive and collect event information;2. **for**
T≥0
**do**   The sensor nodes will conduct simple integration of the data set monitored in the t period, remove redundant and identical data;   **while**
$D_{T_{i}}\neq D_{T_{i-1}}$**do**     The integrated data $D_{T_{i}}$ will be transmitted to its cluster- head sink node;   **end while**3. **end for**.4. The sink node will integrate and standardize the data collected after the *T* period;5. The sink node obtains the weight of each attribute value according to the data fusion algorithm, and synthesizes the final value of each attribute value according to the attribute value;6. The sink node merges the final value of each attribute value into the final prediction result $R_{final}$;7. **If**
$R_{final}$ are changed, **do**   The sink node will periodically transmit the calculated final prediction results to the base station.8. **Else**
   $R_{final}$ will not be transmitted;9. **end if**10. The base station will comprehensively evaluate the collected results to determine the possibility of the event happening $P_{final}$ in the monitoring area;11. **If**
$P_{final}$ are changed, do $P_{final}$ is transmitted to the terminal;12. **Else**
13.   $P_{final}$ will not be transmitted.14. **end if**

Algorithm 1 not only greatly improves the reliability of the entire network, but also fundamentally improves the network life cycle. The main reasons for this are as follows: The base station dispatches a large number of data fusion tasks to each sink node, so that complex data fusion tasks are carried out simultaneously by multiple sink nodes. The base station receives the final prediction results of each cluster, which greatly saves time and improves the real-time performance of the system as a whole, and makes the prediction results available in a timely manner for the terminal staff. In general, wireless sensor network systems are deployed in a large monitoring area, and the predicted events are likely to occur in a certain area. The regional monitoring mentioned in this paper makes it easier to find out which area has the highest probability of occurrence of an event in a timely manner (such as when a portion of the forest is likely to catch fire), improves the accuracy of the prediction system, and enables people to take more directional defense measures. Finally, because the time period of sensor nodes and sink nodes is reasonably allocated in this paper, the communication times of the whole network are reduced so as to extend the life cycle of the whole network.

### 4.3. Case Study: Implementation of Data Fusion Algorithm in Forest Fire Monitoring and Warning System

This paper illustrates the superiority of this data fusion algorithm through the forest fire monitoring and early warning system. In the forest fire monitoring and early warning system, it is usually necessary to monitor the temperature, humidity, smoke concentration and other parameters at various points in the forest. As shown in Figure 2, the forest fire monitoring and early warning system consists of three subsystems: data acquisition subsystem, control center subsystem, and emergency response subsystem. Among them, the data acquisition subsystem mainly uses four types of sensors to perceive and collect data to further predict the possibility of forest fire. They are temperature, smoke concentration, air humidity, and far infrared flame sensors. The first three sensors are used for forest fire prediction, while the far-infrared flame sensor can detect the occurrence of fire and the location and distance of the fire source. The control center subsystem is composed of sink nodes, base stations, and terminals. The sink node first merges the data collected in the cluster to obtain local decisions. Then, the local decision is sent to the base station, and the base station obtains the fire prediction report through the comprehensive analysis of each local decision collected. It can also obtain the fire monitoring report and fire spread prediction based on the changes of the prediction results of multiple cycles, and then send the prediction report to the terminal. The emergency response subsystem is composed of terminals and fire prevention personnel. When the fire prevention personnel see the dynamic information of real-time change prediction, they will formulate relevant emergency plans or measures to prevent forest fire. This paper mainly focuses on the first two subsystems, whose working principles are described as follows:

The set of monitored features may consist of the following:

$F_{1}:$ temperature

$F_{2}$: smoke density

$F_{3}:$ air humidity

$F_{4}$: infrared intensity

The set decisions:

$d_{1}:$ fire unlikely

$d_{2}$: fire likely

$d_{3}$: fire is going on

A decision matrix can be defined as
(9)$$\left[\begin{array}{c} \begin{array}{cc} ↓80 & ↑180 \end{array}\\ \begin{array}{cc} ↓10 & ↑80 \end{array}\\ \begin{array}{c} \begin{array}{cc} ↑65 & ↓25\\ ↑700 & ↓1000 \end{array} \end{array} \end{array}\right]$$
where ↓x indicates any value smaller than *x* that is replaced with *x* and ↑x indicates any value larger than *x* that is replaced with *x*. The above example decision matrix indicates that if feature $F_{1}$ (temperature) is 80 (degrees Fahrenheit) or less, feature $F_{2}$ (smoke density) is 10 or less, and feature $F_{3}$ (humidity) is greater or equal to 65, then decision $d_{1}$ is taken (i.e., fire unlikely). When the sensor node receives such data, it will not transfer the packet to the sink node because there is no possibility of fire, but if feature $F_{1}$ (temperature) is 180 (degrees Fahrenheit) or more, feature $F_{2}$ (smoke density) is 80 or more, and feature $F_{3}$ (humidity) is 25 or less, then decision $d_{2}$ is taken (i.e., Fire likely) and sensor nodes will periodically transmit such data collected to sink nodes. When feature $F_{4}$ indicates that the monitoring infrared ray wavelength is between 700nm-1000nm, decision $d_{3}$ is taken. To prove that the fire has occurred, the sensor node sends the fire occurrence warning packets to its sink node. If the sink node receives several fire occurrence warning packets (the number received is less than 20% of the number of infrared sensor nodes), it means that the probability of fire is low and the sink node will not immediately send the fire warning packet to the base station. If the sink node receives the warning packets of multiple nodes (more than 75%) in the cluster, it will immediately transmit the message to the base station for subsequent remedial measures. In general, such sensor nodes will not send data to cluster head nodes when fire does not occur, so the prediction of fire mainly depends on feature $F_{1},F_{2},F_{3}$. The following data fusion algorithm is used to predict the probability of forest fire:

Step1: In the period *T*, a sink node collects packets from sensor nodes in its cluster. First, we need to filter the data collected for each attribute value. We prefer the data that fire may occur, but considering the accuracy of the data, we keep the data that appear in a certain data interval more frequently and the data that are closer to the possibility of fire. In order to reduce data redundancy and improve data feasibility, we verify whether the following two situations occur together: (1) The data change trend of the same sensor node stays the same and (2) the set of data significantly changes during the entire course. If so, we believe the latter set of data is more reliable and abandon the first set. The process of data standardization is as follows: divide the three attribute values into ten parts according to their interval, and assign values corresponding to 0–1.0. In this way, the collected data sets can be converted into hesitant fuzzy sets. That is, for the attribute $F_{1}$, its corresponding value interval is normalized to {[80,90]→[0,0.1]},{(90,100]→(0.1,0.2]},⋯,{(170,180]→(0.9,1.0]}. The value interval corresponding to the attribute $F_{2}$ is normalized to {[10,17]→[0,0.1]},{(17,24]→(0.1,0.2]},⋯{(73,80]→(0.9,1.0]}. The value interval corresponding to the attribute $F_{3}$ is normalized to {[65,61]→[0,0.1]},{(61,57]→(0.1,0.2]},⋯{(29,25]→(0.9,1.0]}. In order to illustrate the problem more concisely, we simplify the integrated data and assume that each attribute value corresponds to five sets of data, as shown in Table 1.

Table 2 is the corresponding hesitant fuzzy sets matrix after the standardization of Table 1. As shown in Table 1, the monitoring values can be three, two, or one, indicating that when the sensors collects data, they might monitor three, two, or one data value in a collection cycle. The monitoring values of different numbers also reflect that the attribute values of some sensor nodes monitoring areas are variable, while some are not. 

Step2: Formula (5) is used to transform the hesitant fuzzy set matrix into the entropy matrix, as shown in Table 3.

Step3: Formula (6) is used to calculate the weight of each type of sensor node:
(10)$$w={\left(0.2934,0.1777,0.2521,0.1338,0.143\right)}^{T}$$

Step4: According to Formula (7), the final data values of all the measured attribute values are synthesized:$${\bar{D}}_{F_{1}}=155.8,{\bar{D}}_{F_{2}}=25.5,{\bar{D}}_{F_{3}}=46.4$$

Step5: Considering that air humidity is the most critical factor leading to forest fire, the weight vector of these three attribute values given by the system is $w={\left(0.3,0.2,0.5\right)}^{T}$. Then, Formula (8) is used to calculate the possibility of fire in this cluster as follows:(11)$$P=0.3\times \frac{155.8}{180}+0.2\times \frac{25.5}{80}+0.5\times 1-\frac{46.5}{65}=47\%$$

Step 6: The sink node will send the synthetic attribute values in steps 4 and 5, together with the 47% probability of fire prediction in the cluster, to the base station. The base station will make a comprehensive comparison based on the results sent by each sink node, so that it can easily predict which region has the highest probability of fire. At the same time, it can also judge the dynamic change trend of a certain region according to several consecutive cycles, in order to prepare for the follow-up work.

This example is mainly used to illustrate the operation idea of this algorithm, so it simplifies the data. The filtered data in reality is much more complex than the filtered data in the example, so that the accuracy rate is higher. At the same time, we can see from this example that if some sensor nodes in the cluster do not work or die, as long as the monitoring range of similar surviving sensor nodes in the cluster can cover the whole cluster, the system still operates normally, thus improving the robustness of the whole network.

## 5. Evaluation and Discussion

In this section, we introduce the performance analysis of the algorithm and compare it with two other algorithms. In the experiment, we mainly focus on the influence of this algorithm on the total energy consumption of the network and the overall network life.

### 5.1. Experiment Setup

In order to explain the advantages of this algorithm more clearly, we choose a forest fire prediction system to exemplify the proposed algorithm. This paper mainly uses omnet++ for experimental simulation, and some data are analyzed and explained by MATLAB. In this simulation experiment, the monitoring area of the experimental simulation is 50 m × 50 m, in which a base station is placed. Its energy calculation and storage capacity are sufficient, and the communication radius is 100 m. It is the destination node for all sink nodes to send data. A total of 30 sink nodes are evenly distributed to the monitoring area, they have sufficient energy and processing capacity, the monitoring area is based on its own as the center, and the communication radius is an *R* = 15 m round area, so the sink nodes can cover the entire monitoring area. As discussed earlier, the sink nodes collect data from their neighboring sensor nodes and use the data fusion algorithm to make local decisions which are then sent to the base station. There are 150 sensors in each of the four sensors of sensing temperature, air humidity, smoke concentration, and far infrared ray distributed in the monitoring area. The communication radius of the sensor is *r* = 5 m, and the sensor mainly induces its relative event attribute value. Besides, sensor nodes can be used as intermediate nodes for data transmission. In order to further prove the rationality of node density setting in the experiment, we specially summarized the reasonable range of node density. As mentioned in [25], the optimal density of nodes is ρ=1.03. In [26], this study proposed increasing the node density near the base station during deployment to compensate for the requirement of high energy consumption. The results show that the optimal node density distribution is 4:2.5:1, and the closer it is to the base station, the greater the sensor node density. In [27], it mainly studies the relationship between node density and the number of hops. Experimental results show that when the node density is 5, it is most beneficial to network data transmission. Therefore, the node density setting is practical in our experiment, which will not lead to the redundancy of data collection due to the excessive node density. In addition to the different event attributes induced by these sensor nodes, other performance parameters are all the same, as shown in Table 4. 

In this simulation system, sink nodes and sensor nodes periodically transmit data to their parent nodes. The sink nodes report to the base station at period *T* = 100 *s*, and the sensor nodes report data to the sink nodes at period *t* = 20 s. The sensor sampling rate is the response speed of the sensor node to the signal, and its value is equal to the reciprocal of the response time. In our simulation system, the sensor sampling rate value is Δf=0.5 S/s. 

In addition, the event generator aims to simulate the forest fire scene in this paper. It does the simulation by releasing four parameters: temperature, air humidity, smoke density, and far infrared ray. In order to test the feasibility of simulation, we must give a reasonable value range for these four parameters: temperature range is 0–300 °C, as related data show that wood fires are 180–290 °C. If the temperature is detected in this range, it is considered that the fire is likely to happen. The air humidity is the absolute air humidity and the value is 0–100. When the value of the air humidity is 25, it is considered that fire is very likely to happen, but when the air humidity value is more than 60, fire is unlikely to happen. The smoke concentration range is 0–100 and the infrared wavelength is 700–1000 nm. 

In this simulation experiment, the infrared signal cannot be released arbitrarily. The constraints are that when the temperature is above 200 °C and humidity of the air is less than 40, the event generator can send out an infrared signal. Otherwise, it cannot send out an infrared signal. In addition, the smoke concentration should not be higher than 25 in the absence of fire. The event generator for a period will be within each cluster more $t^{\prime }=5\;s$ to send an event signal to imitate the possibility of a fire. When the released event signal is certain to avoid a fire, the corresponding sensor node will not receive the signal. Only when the received data is in the range of possible fire will the sensor node collect the data (see the decision table above for forest fire occurrence) and then transmit it to the sink node. We assume that the data published by the event generator is in an 80% probability range of fire occurrence, and in this 80% probability, the probability of fire occurrence is 20%.

### 5.2. Result and Discussion

This paper is compared with two algorithms, namely a Data Fusion Method in Wireless Sensor Networks (DFM) [14] and Data Fusion in Wireless Sensor Networks using Fuzzy Set Theory (DFFST) [15]. The reason for choosing these two algorithms is that both choose fuzzy theory in the data fusion of a wireless sensor network, and they carry out forest fire prediction analysis, which is very similar to our algorithm. DFM mainly aims at improving the service quality of the network and reducing the energy consumption of a wireless sensor network, and proposes a data fusion method based on fuzzy theory. However, this method is only used to distinguish and aggregate the authenticity of the collected data, and does not further investigate, so it is far from the basic performance of this paper. The DFFST algorithm mainly uses the fuzzy set theory to aggregate the data of multiple sensors at the cluster level. To some extent, it can deal with the inaccuracy and conflict inherent in environmental data reading. However, in reality, if we do not filter the collected data before processing it, it is likely that a large amount of false information will be adopted. At the same time, the accuracy of this algorithm cannot be guaranteed. It sets the data value of the same interval collected as the median value of the interval, which is convenient for processing, but actually greatly reduces the accuracy. In order to compare the experimental rigor, the experimental configuration of these three algorithms is the same.

In this paper, the total number of surviving nodes, residual total energy, and average time delay of the three algorithms are compared with the increase of network operation times. As shown in Figure 3, it mainly reflects the change in the number of remaining surviving sensor nodes in the network with the increase of network operation times. The total numbers of sensor nodes are 600 in the three algorithms, and the nodes are evenly divided according to the types of nodes in the algorithm. As shown in Figure 3, when the algorithm runs to 550 rounds, the number of sensor nodes will suddenly decrease to 150. This is mainly because the sensor sensing air humidity, smoke concentration, and temperature in this experiment dies, while the node that continues to survive is the far infrared sensor node, which is mainly related to the fire probability set in this experiment. However, DFFST and DFM algorithms are distributed in the network running times to 200 and 150 times, and nodes start to die in succession. It can be said that this algorithm is superior to the above two algorithms in the uniform consumption of network energy. The main reason why the data fusion algorithm can achieve this effect is that the sensor nodes periodically simply integrate the collected data and transfer non-repeated data to the sink nodes. In this way, both the number of data transmission times and the transmission of redundant data are reduced, and the energy consumption of each sensor node is greatly reduced. At the same time, the selection of redundant data by sink nodes and base stations is reduced and the efficiency of data fusion is improved. Therefore, the survival time of sensor nodes is enhanced, and the life cycle of the whole network is prolonged. The subsequent surviving far-infrared flame sensor nodes can only detect the occurrence of fire, so they will gradually die with the increase of network operation times.

Figure 4 mainly shows the relationship between the total residual energy of the network and the number of network runs. It can be clearly seen from Figure 4 that the total residual energy of sensor nodes of the proposed data fusion algorithm is significantly higher than that of the other two algorithms. Moreover, the total energy consumption of the proposed algorithm is linearly related to the network operation times. It is explained that the proposed algorithm can save energy and ensure the stability of the whole network energy consumption. The main purpose of the proposed algorithm is to reduce the transmission of redundant data and thus reduce most energy consumption. In addition, data fusion is completed by the sink node. The sink node (cluster head) in this paper is fixed, so the energy consumption of selecting the cluster head is reduced. In the later period of network operation, only some sensor nodes are working due to the death of sensor nodes, so the overall energy consumption decreases.

Figure 5 mainly compares the relationship between network operation times and average time delay. The average time delay referred to in this paper is the transmission time of packets and data processing time in the network, excluding the waiting time of packets at the sink node. It only counts the transmission time delay of packets from the sensor node to the sink node, and then from the sink node to the base station. It can be clearly seen from Figure 5 that the average time delay of the proposed algorithm is significantly lower than that of the other two algorithms. The main reason for this is that the data packets transmitted by the proposed algorithm are relatively small after fusion. Each sensor node delivers data packets to the sink node on a periodic basis, which reduces network congestion. In addition, this paper makes local decisions at the sink node, which can greatly reduce the amount of data in the network. At the same time, the base station can also produce the prediction results quickly, which improves the real-time performance of the whole system and makes the system more suitable for application scenarios with a strong real-time performance. As shown in Figure 5, after 550–600 network runs, the average time delay of DFFST and DFM is not applicable due to a large number of dead nodes. Therefore, we leave those histograms empty. From the perspective of the whole network operation cycle, with the increase of network operation times, the average time delay will also increase to different degrees. Because the number of network nodes and the number of intermediate nodes are reduced, the average time delay of the whole network will increase and the overall transmission efficiency of the network will decrease.

## 6. Conclusions and Future Work

This paper demonstrates that data fusion in wireless sensor networks can significantly reduce raw data transmission between networks, which in turn increases the network life and prevents base stations from being saturated. Specifically, we applied hesitant fuzzy entropy to the data fusion of a wireless sensor network, which greatly reduces the amount of data transferred in the network and improves the real-time performance of the entire system. We have shown that the proposed data fusion algorithm could improve the reliability of data compared to the state-of-the-art. In the future, we plan to integrate the proposed data fusion algorithm into other system research areas, such as big data systems [28], key-value stores [29], data compression [30], and blockchains [31].

## Figures and Tables

**Figure 1 sensors-19-00784-f001:**
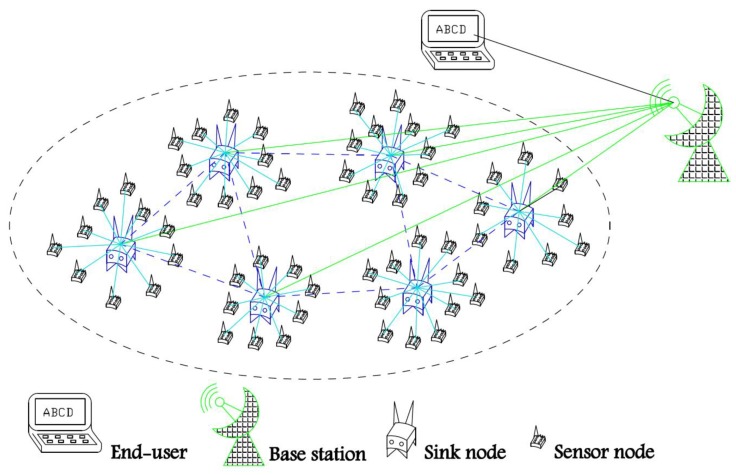
Schematic Diagram of the Network Model.

**Figure 2 sensors-19-00784-f002:**
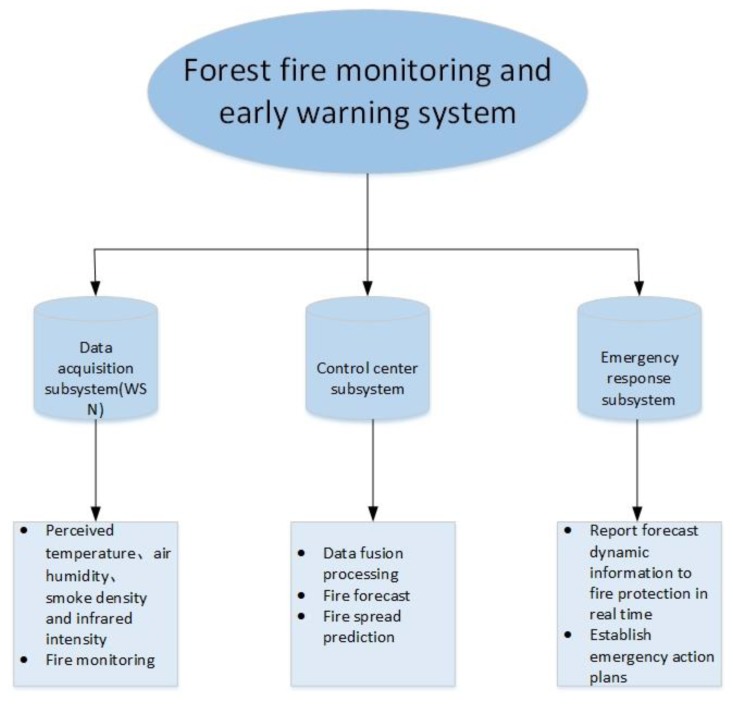
Forest fire monitoring and early warning system.

**Figure 3 sensors-19-00784-f003:**
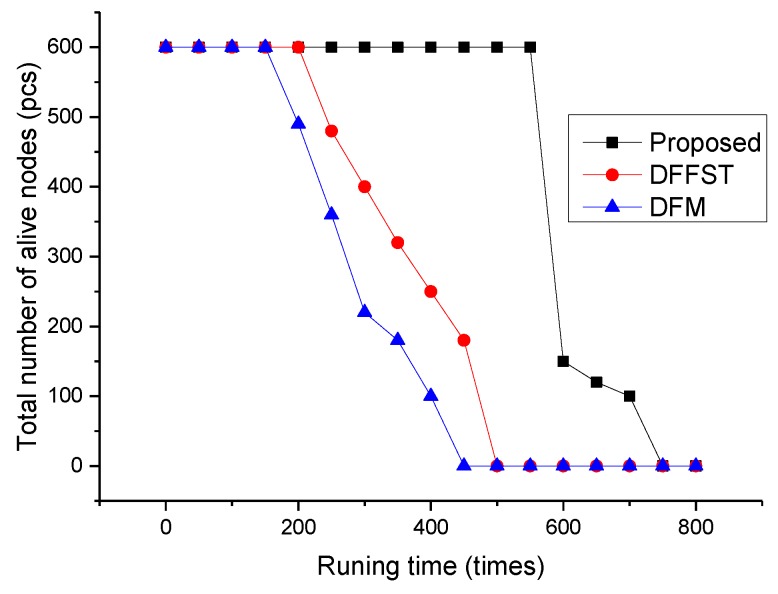
Relationship between the number of network runs and the total number of surviving sensor nodes.

**Figure 4 sensors-19-00784-f004:**
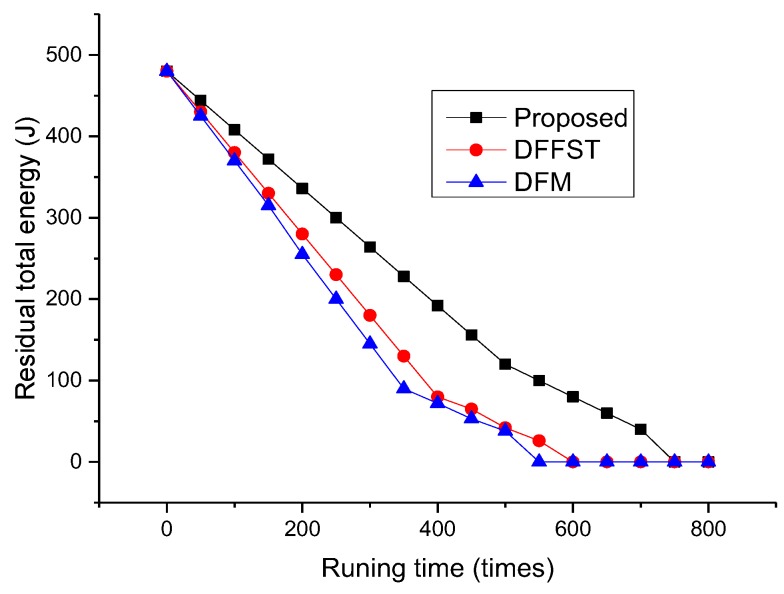
Relationship between the total number of network runs and the remaining total energy.

**Figure 5 sensors-19-00784-f005:**
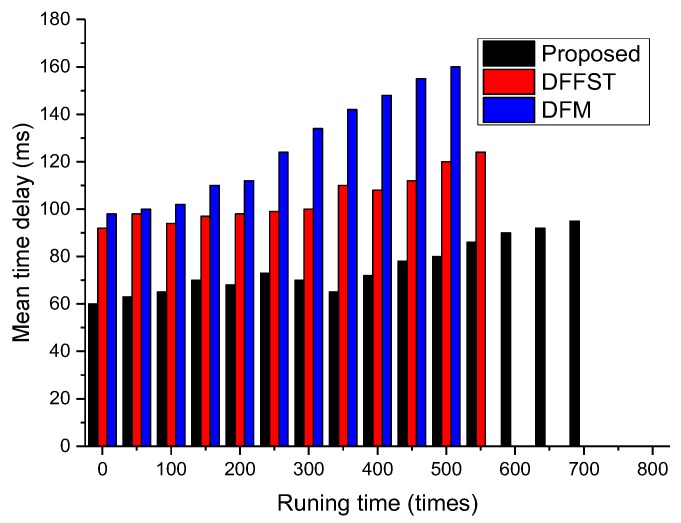
Relationship between the total number of network runs and the average time delay.

**Table 1 sensors-19-00784-t001:** The actual measured value after screening.

	$F_{1}$	$F_{2}$	$F_{3}$
$S_{1}$	{150,155,160}	{15,18,24}	{45,50,57}
$S_{2}$	{155,160}	{31}	{35,42,45}
$S_{3}$	{160,165,169}	{24,30}	{40,45,50}
$S_{4}$	{150,155}	{24,32,38}	{49}
$S_{5}$	{140,145}	{24}	{41,45,48}

**Table 2 sensors-19-00784-t002:** Hesitant fuzzy decision matrix.

	$F_{1}$	$F_{2}$	$F_{3}$
$S_{1}$	{0.7,0.75,0.8}	{0.04,0.17,0.2}	{0.2,0.38,0.5}
$S_{2}$	{0.75,0.8}	{0.3}	{0.5,0.63,0.75}
$S_{3}$	{0.8,0.85,0.89}	{0.2,0.29}	{0.38,0.5,0.58}
$S_{4}$	{0.7,0.75}	{0.2,0.31,0.4}	{0.4}
$S_{5}$	{0.6,0.65}	{0.2}	{0.43,0.5,0.6}

**Table 3 sensors-19-00784-t003:** Entropy matrix.

	$F_{1}$	$F_{2}$	$F_{3}$
$S_{1}$	0.7580	0.4782	0.9227
$S_{2}$	0.7068	0.8460	0.9378
$S_{3}$	0.5300	0.7480	0.9990
$S_{4}$	0.8046	0.8499	0.9617
$S_{5}$	0.9401	0.6505	0.9994

**Table 4 sensors-19-00784-t004:** Additional parameters in evaluations.

Notion	Description	Value
$E_{elec}$	Transmitter orreceiver energy consumption	50 nJ/bit
$\epsilon_{amp}$	Amplifier energy consumption	100 pJ/bit$/m^{2}$
C(trans)	Transmitter power	0.020 w
C(rec)	Receiver power	0.029 w
E(init)	Initial energy	0.8 J
$E_{data}$	Data fusion energy consumption	5 pJ/bit
MAC	Media Access Control	IEEE 802.15.4
*k*	Transmission attenuation	4
$E_{communication}$	Built-in communication equipment	Zigbee

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
