# Peer review of "A New Data Fusion Algorithm for Wireless Sensor Networks Inspired by Hesitant Fuzzy Entropy"

_sensors, 2019, doi:10.3390/s19040784_

Round 1
Reviewer 1 Report
The paper that is presented has merit and introduces a new approach to the problems that can occur in wireless sensor networks (WSN). The use of Hesitant Fuzzy Entropy (HFE) to the consumption of energy in WSN is novel. The results that are presented demonstrate an understanding of the scientific process and use a prime use case to demonstrate this. The comparison to the other algorithms shows that the proposed approach has strengths and out performs them across three measures.
There are, however, some issues that need to be addressed. Within the Abstract (line 17), the new algorithm is proposed to improve data accuracy. This is not discussed further in the paper. It also can be said that the use of some of the techniques within the algorithm could impact reliability or maintain the current reliability. In line 369 – 373, a discussion is had in regard to large changes in the data supplied by a node as having higher reliability in the case of recognition of an event. This is impart can be true, however large changes (for instance consistent values changing to low) may be caused by the degradation of the sensor (battery, damage etc). The data may not be seen to be reliable as a result. This needs to be acknowledged and discussed or incorporated into the algorithm.
Additionally, the experimental results do not compare other algorithms on this parameter, but focus on energy consumption and network life. Again, this needs to be addressed.
It can be asked also of the local decision making was taken into account for the comparison with other algorithms as this does not appear to be shown.
Aside from the above, there are a number of grammatical and structure errors:
Line 46 – 48 -> This section is difficult to follow.
Line 73 – 74 -> This section is difficult to follow.
Figure 1 – Needs revision as difficult to follow and is not clear (screen grab?)
Algorithm 1 – Consider indents to help the reader to process.
Line 435 – 447 -> There are many elements in here that are confused. This needs revision.
Author Response
Response to Reviewer 1 Comments
Point 1: Within the Abstract (line 17), the new algorithm is proposed to improve data accuracy. This is not discussed further in the paper. It also can be said that the use of some of the techniques within the algorithm could impact reliability or maintain the current reliability.
Response 1: We agree with your comment: the reliability is the focus of this paper and we have revised the abstract and other places in the text. We would like to elaborate a bit more about how accuracy is also benefited from our work. In this paper, the data transmission is reduced by removing redundant data to meet the requirements of improving the system’s performance. For the same attribute value of an event, there will be many sensor nodes perceiving the value, and these sensor nodes will have different or distorted perception of the attribute value due to various factors in the environment (such as obstacles, noise, etc.) or their own reasons (such as power consumption, etc.). So, if we might be able to obtain the real value but in most cases, the monitored values are only estimates of the real value. Because the perceived value deviates from the real value, the proposed algorithm of this paper aims to eliminate some of the redundant data. Many of these data are far from the true values. This paper argues that if the monitored value is within a numerical interval to the real value, then we believe they are good estimates to the real value; otherwise, the data would be considered invalid and would be removed, which improves the reliability of the monitored values. Then, we apply hesitant fuzzy entropy to value synthesis between the area and calculate the final value to prepare for subsequent synthesis of local decisions.
Point 2: In line 388 – 390, a discussion is had in regard to large changes in the data supplied by a node as having higher reliability in the case of recognition of an event. This is impart can be true, however large changes (for instance consistent values changing to low) may be caused by the degradation of the sensor (battery, damage etc). The data may not be seen to be reliable as a result. This needs to be acknowledged and discussed or incorporated into the algorithm. Additionally, the experimental results do not compare other algorithms on this parameter, but focus on energy consumption and network life. Again, this needs to be addressed. It can be asked also of the local decision making was taken into account for the comparison with other algorithms as this does not appear to be shown.
Response 2: Thank you very much for your feedback. We agree that the description in the article can be improved. The description has been revised in lines 386-390. If the sensor node's perceived data significantly change due to the real change of the monitored object's attribute value, then the value of the sensor nodes in the monitoring area monitoring will follow the same trend of change (e.g., individual sensor node’s value distortion caused by battery monitoring). This paper will delete the redundant data when the primary copy is deleted because it is not considered accurate. Therefore, it will not affect the monitoring results. If most of the sensor nodes in the whole system experience energy failures, the sensor network should be abandoned or rearranged. This article proposes a new data fusion algorithm for improving the performance of the whole network. Since network performance is one of the most impactful factors to reducing the energy consumption and prolonging the life cycle of the whole network, this article does not discuss other parameters in much detail.
Point 3: Grammar problem: Line 45 – 49 -> This section is difficult to follow.
Response 3: Fixed.
Point 4: Grammar problem: Line 72 – 76 -> This section is difficult to follow.
Response 4: Fixed.
Point 5: Figure 1 – Needs revision as difficult to follow and is not clear (screen grab?)
Response 5: We redraw the figure with higher resolution.
Point 6: Algorithm 1 – Consider indents to help the reader to process.
Response 6:Fixed.
Point 7:Line 465-481-> There are many elements in here that are confused. This needs revision.
Response 7: We have broken down the original paragraph into four paragraphs. We also explain each of the parameters in much more detail (lines 434-481).

Reviewer 2 Report
The proposal seems to be interesting, but there are several aspects which are not clear that lead to a negative impression regarding the strength to the proposal. The are assumptions (most related with the communication aspects) that must be clarified and explained with detail. The paper must be improved to present a sound proposal.
pg 4, ln 148. In the regular WSN terminology, the network has only a sink node, which is different from a cluster head. Cluster heads, usually, acts as routers conveying information to the sink node. In this case, the network topology is similar to a tree.
The network model presented is, in my opinion, is much closer to a group of individual WSNs, each one with a sink node in a centralized topology, where each sink node communicates with a base station.
Ln.158. how can one node know the (residual) energy of the others?
Ln 167. Should not exist a processing state in which the node is acquiring and processing data ? These operations could take more time and energy than transmitting (eg. AD conversion, computing fuzzy sets, etc)
Ln 174. Please explain “by one square meter per bit”. Is the energy to transmit a bit to radius of 1 square meter?
Ln 175. Please provide some references for the ‘k’ (attenuation index)
Ln 185. Explain the statement: “when a sensor node have received hello message packets from different sink nodes, it will automatically perceive and calculate the distance between the sink node and itself, and then select the nearest sink node as the cluster head”. How can a node compute the distance to the other nodes ? Based on what? This a complex task that could be affected by many factors (eg. attenuation, terrain, interferences). Using RSSI ? (“Is RSSI a Good Choice for Localization in Wireless Sensor Network?”, Karel Heurtefeux ; Fabrice Valois).
Lns. 190-197. Has a sink node the capability to communicate directly with the base station? This is not clear from the text. If true, than you have a problem: how to justify that the farthest sink has the capability (energy, radio power etc) to communicate with the base station (assume a km range network). If it has this capability why it should communicate with others sinks ? If negative, how the information coming from sink nodes reaches to the base station? (e.g. all data goes through a single sink that is closer to the base station ?)
Lns. 190-197. You must explain more carefully how the network formation and data transmission is made. I understand that this is not the paper purpose, but this is, in my opinion, necessary to have a sound proposal (i.e. which are the mechanisms/proposal/protocols already available on the literature which can be used to guarantee that the network operates in this manner). Honestly, many of these proposals are, in my opinion, “wishful thinking” assumptions. I will give you a few examples:
-- what happens when a sink fails? There is any network reconfiguration? What happens to that cluster?
-- since base station is mobile, the network must employ a highly dynamical protocol to guarantee that the information. How it operates?
Ln 296. Please explain the meaning of “arithmetic unit”. Has the sink node 2 processors, one for running “regular applications” (eg. transmitting, etc.) and other for “math operations” (eg. data fusion) ? This is quite strange.
Ln 361” proves that the message reliability is relatively high“. How this prove is made?
Ln. 453. Table 4. Nodes use WiFi to communicate ? This is not a credible assumption. First, Wifi is energy hungry, so use this kind of communication would lead to a quick energy depletion (data presented for energy reception, transmission is not for wifi certainly). Second, Wifi has 100m communication range, which is not according to the sink and sensor ranges (unless you have some way to reduce the transmission power).
The number of nodes vs. the covered area seems to be unrealistic: 600 nodes in 2500 sqrm (1 node in 4 sqrm). This leads certainly to very optimistic results since the redundancy degree is very high. Since you are using a simulation approach to validate the proposal, it costs nothing to present a more credibly scenario. In a forest fire context, it would be reasonable to present scenarios of several Km with hundreds of nodes.
Ln 477. “550 rounds”. A round is 100s or 20s ? Either way, in the worst case after 15 hours the network has already lost 75% of the nodes….for a forest fire detection solution this is not a good result.
L 512 “excluding the residence time of packets at the sink node”. Why did you exclude this component? This is important to measure the network reaction time (in this context it is useless to have a network with long delays which prevents a quick fire detection). The results presented aren’t fair.
Ln 522. They are not zero, but infinite (that is why they are not shown in the figure)
Author Response
Response to Reviewer 2 Comments
Point 1: pg 4, ln 143. In the regular WSN terminology, the network has only a sink node, which is different from a cluster head. Cluster heads, usually, acts as routers conveying information to the sink node. In this case, the network topology is similar to a tree. The network model presented is, in my opinion, is much closer to a group of individual WSNs, each one with a sink node in a centralized topology, where each sink node communicates with a base station.
Response 1: Thanks for your feedback. In general, a cluster head is generated by an election of sensor nodes. The focus of this paper is to study data fusion, so the cluster head is replaced by the sink node in the proposed approach. In this way, as the sink node has high reliability, we did not consider the case where the cluster head is frequently changed.
Point 2:Ln.160. how can one node know the (residual) energy of the others?
Response 2: Thank you for your comment, very good point. This work is based on our prior work, and some assumptions should have been spelt out more explicitly. In Lines 160-162, we have addressed this in the revised manuscript. When a node sends a message, the remaining energy information of the node will be carried in the header file of the message such that neighbor nodes are aware of the remaining energy of other nodes.
Point 3:Ln 172-173. Should not exist a processing state in which the node is acquiring and processing data? These operations could take more time and energy than transmitting (eg. AD conversion, computing fuzzy sets, etc)
Response 3: Since the data fusion and other large computing tasks in this paper are completed on the sink node, and the power set in the sink node is sufficient, we did not emphasize the power of the sink node. In Section 3.2, the energy model is for sensor nodes. In this article, the sensor nodes’ energy is limited and they are only responsible for sensing and forwarding data. That is, they do not add processing states. Moreover, even if sensor nodes carry out simple calculation, their energy consumption ratio in calculation and communication is 1:3000 (Line 45-48). Therefore, the energy consumption for sensor nodes to conduct simple calculation (such as deleting duplicate data) can be ignored. We have revised the manuscript accordingly (Lines 171-175).
Point 4:Ln 178. Please explain “by one square meter per bit”. Is the energy to transmit a bit to radius of 1 square meter?
Response 4: represents the energy consumption per bit transmitted by the transmitting amplifier per square meter, which is the energy consumption unit of the transmitting amplifier. The communication of sensor nodes is completed by the wireless transceiver. is related to the energy consumption of information transmitted by sensor nodes, but different from the energy consumption of receiving messages.
Point 5:Ln 180. Please provide some references for the ‘k’ (attenuation index)
Response 5: Reference has been added as [4].
Point 6:Ln 186. Explain the statement: “when a sensor node have received hello message packets from different sink nodes, it will automatically perceive and calculate the distance between the sink node and itself, and then select the nearest sink node as the cluster head”. How can a node compute the distance to the other nodes? Based on what? This a complex task that could be affected by many factors (e.g. attenuation, terrain, interferences). Using RSSI ? (“Is RSSI a Good Choice for Localization in Wireless Sensor Network?”, Karel Heurtefeux ; Fabrice Valois).
Response 6: In this paper, RSSI is used for the calculation of perceived ability, because the focus of this paper is the new data fusion challenge, we did not elaborate RSSI. We have provided a reference to a more detailed discussion on RSSI, in reference [4], at lines 165-166.
Point 7:Lns. 207-209. Has a sink node the capability to communicate directly with the base station? This is not clear from the text. If true, than you have a problem: how to justify that the farthest sink has the capability (energy, radio power etc) to communicate with the base station (assume a km range network). If it has this capability why it should communicate with others sinks ? If negative, how the information coming from sink nodes reaches to the base station? (e.g. all data goes through a single sink that is closer to the base station ?)
Response 7: In this paper, the base station is able to communicate with all sink nodes directly, but this is not the case between sink nodes. If the base station is within the communication range of a sink node, that sink node can communicate directly with the base station; otherwise, it has to transmit messages through multi-hop, in which the intermediate node can be both sink nodes and sensor nodes. We have revised the manuscript at lines 207-212.
Point 8: Lns. 190-197. You must explain more carefully how the network formation and data transmission is made. I understand that this is not the paper purpose, but this is, in my opinion, necessary to have a sound proposal (i.e. which are the mechanisms/proposal/protocols already available on the literature which can be used to guarantee that the network operates in this manner). Honestly, many of these proposals are, in my opinion, “wishful thinking” assumptions. I will give you a few examples:
-- what happens when a sink fails? There is any network reconfiguration? What happens to that cluster?
-- since base station is mobile, the network must employ a highly dynamical protocol to guarantee that the information. How it operates?
Response 8: Thanks for pointing that out. We have elaborated that part in the revised manuscript at Lines184-207. In addition, the revised manuscript provides reference 4 for those readers who are interested in more details. In essence, the sink node’s energy we set is sufficient, so there is no noticeable sink node failure. If the sensor nodes die in a large area, the wireless sensor network needs to be rebuilt. In this system, the base station mainly collects local decisions from sink nodes and merges the collected local decisions to reach the final decision.
Point 9:Ln 312. Please explain the meaning of “arithmetic unit”. Has the sink node 2 processors, one for running “regular applications” (eg. transmitting, etc.) and other for “math operations” (eg. data fusion) ? This is quite strange.
Response 9: We have clarified this in the revised manuscript. The arithmetic unit is now changed to the data processing unit (Lines 311-313). In general, a sink node or a sensor node is composed of data processing unit, communication unit, sensor and power model. Data processing and data fusion are completed in the data processing unit, while communication is completed in the communication unit.
Point 10:Ln 378” proves that the message reliability is relatively high“. How this prove is made?
Response 10: This part has been revised in the article, see Lines 375-379. The infrared sensors monitor the fire light then send the warnings of fire package to sink nodes. If only a few sensor nodes send such warning packets, sink nodes will not immediately alert the base station. If most infrared sensor nodes send warnings, the sink node will decide whether there is indeed a fire happening.
Point 11:Ln. 484. Table 4. Nodes use WiFi to communicate ? This is not a credible assumption. First, Wifi is energy hungry, so use this kind of communication would lead to a quick energy depletion (data presented for energy reception, transmission is not for wifi certainly). Second, Wifi has 100m communication range, which is not according to the sink and sensor ranges (unless you have some way to reduce the transmission power).
Response 11: Table 4 shows some basic information about the sensor nodes. In this paper, sensor nodes do not use WiFi for communication, but adopt the form of broadcasting. Sensor nodes usually transmit data to the outside world through the built-in communication system. The state-of-the-art wireless communication in the market is Zigbee, WiFi and 433MHz. Zigbee is used in this paper because of its low power consumption, high reliability, strong anti-interference and easy network layout. We have updated Table 4 accordingly.
Point 12:The number of nodes vs. the covered area seems to be unrealistic: 600 nodes in 2500 sqrm (1 node in 4 sqrm). This leads certainly to very optimistic results since the redundancy degree is very high. Since you are using a simulation approach to validate the proposal, it costs nothing to present a more credibly scenario. In a forest fire context, it would be reasonable to present scenarios of several Km with hundreds of nodes.
Response 12: In order to monitor data more accurately, high node density is one of the most important requirements of wireless sensor networks. A sensor node characterized by "tiny" has a low cost, and its basic structure includes microprocessors, limited energy, limited memory, etc. Therefore, we need to exchange data reliability through high-density sensor nodes. In a real forest fire prevention system, in order to improve the reliability of data, the density of sensor nodes will also be high. The node density in this paper is reasonable because in most cases, it is a practical setup to have 1-5 sensor nodes per square meter, Please see Lines 447-456 for detailed description.
Point 13:Ln 507. “550 rounds”. A round is 100s or 20s ? Either way, in the worst case after 15 hours the network has already lost 75% of the nodes….for a forest fire detection solution this is not a good result.
Response 13: 100s and 20s have nothing to do with the number of rounds in the whole system. In this paper, sensor node and sink node do not transmit the data packet to the superior node as soon as they receive it, but in a periodical fashion. Therefore, 100s and 20s in this paper represent the period of a single sink node reporting information to the base station and the period of sensor node transmitting data to the base station, respectively. In order to improve the performance of the whole system, when the collected data changes, the data packet will be transmitted to the superior node periodically. In this paper, we know that the value detected by the sensor node is collected and processed only in the interval when the event is likely to occur. To verify that the system performance, the system event is triggered in , and 80% of these attribute values are in the range where fire may occur(Lines 474-481). That is why in 15 hours the network may lost up to 75% of the nodes. In the real world, fires are unlikely to happen that often, so the monitoring system can run for a much longer time than 15 hours.
Point 14:L 542 “excluding the residence time of packets at the sink node”. Why did you exclude this component? This is important to measure the network reaction time (in this context it is useless to have a network with long delays which prevents a quick fire detection). The results presented aren’t fair.
Response 14: Thank you very much for your comments. In general, the time delay is indeed the propagation time in the network, but this paper does need to include the data processing time because a more accurate description would not include the delay caused by the fact that we set the data to be transmitted periodically. Since the data is integrated on the data processing unit, it will be sent out after the integration. During the time interval (100s/20s) the system waits for more data. Therefore, the time delay in this paper should include the summation of the time of data transmission in the network and data processing time. This part has been revised in the paper (Lines 541-543).
Point 15:Ln 553. They are not zero, but infinite (that is why they are not shown in the figure)
Response 15: Thank you very much for your comment. The description here was not very accurate, and we have revised accordingly, see Lines 552-554. The goal here is to explain that when most sensor nodes in the network are dead, there is no message to be transmitted, and time delay is not applicable. Therefore, it is not entirely correct to either plot it as zero or infinite, and that is why we just leave the space empty.

Round 2
Reviewer 2 Report
The authors have addressed my comments, and the paper has been greatly improved.
However, I’m not fully convinced with the explanation presented for communication layer. Originally, the authors state that nodes use WiFi, and now they correct to Zigbee. However, in table 4 the MAC layer is still IEEE 802.11 when it should be IEEE 802.15.4. Although this aspect doesn’t reduce the merit of the proposal, it should be clarified in the paper: which MAC layer the nodes use to communicate?
Author Response
Response to Reviewer 2 Comments
Point 1:However, I’m not fully convinced with the explanation presented for communication layer. Originally, the authors state that nodes use WiFi, and now they correct to Zigbee. However, in table 4 the MAC layer is still IEEE 802.11 when it should be IEEE 802.15.4. Although this aspect doesn’t reduce the merit of the proposal, it should be clarified in the paper: which MAC layer the nodes use to communicate?
Response 1: Thank you very much for your comments. IEEE 802.15.4 is the IEEE specification at the physical layer and the media access control (MAC) layer for low-speed wireless personal domain networks (LR-WPAN). The protocol can support simple devices that consume the limited power and generally operate in a personal activity space (10-meter diameter or less). The protocol supports two network topologies, namely, single-hop star or multi-hop peer topologies when the communication line exceeds 10 meters. The MAC layer of wireless sensor nodes aims to save energy and achieve autonomy. We agree with the reviewer that IEEE 802.15.4 is more appropriate for wireless sensor networks. We have updated Table 4 accordingly in the revised manuscript.
